# Improved Stability of Blue Colour of Anthocyanins from *Lycium ruthenicum* Murr. Based on Copigmentation

**DOI:** 10.3390/molecules27186089

**Published:** 2022-09-18

**Authors:** Kai Deng, Jian Ouyang, Na Hu, Qi Dong, Chao Chen, Honglun Wang

**Affiliations:** 1Qinghai Provincial Key Laboratory of Tibetan Medicine Research, CAS Key Laboratory of Tibetan Medicine Research, Northwest Institute of Plateau Biology, Xining 810008, China; 2University of Chinese Academy of Sciences, Beijing 100049, China; 3Huzhou China–Science Innovation Centre of Plateau Biology, Huzhou 313000, China; 4Beijing Tongrentang Health Pharmaceutical (Qinghai) Co., Ltd., Haixi 817000, China

**Keywords:** *Lycium ruthenicum* Murr., anthocyanins, copigmentation, blue colour, stability

## Abstract

Natural blue food colourant is rare. The aim of this work was to screen compounds from the common copigments that could improve the blue tones of anthocyanins (ACNs) and to investigate the effect of different copigments on the colour stability of anthocyanins in neutral species. International Commission on Illumination (CIE) colour space, UV, IR, NMR, atomic force microscopy (AFM) and computational chemistry methods were utilised to evaluate ACNs from *Lycium ruthenicum* Murr. (*LR*), which is complexed with food additives and biological agents. The results indicate that Pro−Xylane (PX), Ectoin (ECT) and dipotassium glycyrrhizinate (DG) enhance the blue colour of the ACNs. ACNs−PX presents a colour close to Oxford Blue and has a surface height of 2.13 ± 0.14 nm and slightly improved stability. The half−life of ACNs−DG is improved 24.5−fold and had the highest complexation energy (−50.63/49.15) kcal/mol, indicating hydrogen bonds and π−π stacking forces enhance stability. These findings offer a new perspective for anthocyanin utilisation as a blue colourant and contribute to the large−scale application of *LR*.

## 1. Introduction

The colour of foods is essential for attracting customers. The blue colour is noted for its novelty and can attract the attention of young customers [1,2]. Due to the rareness of blue colourants in nature, blue colourants in food application are predominantly synthetic, such as Patent Blue, Indigo Carmine and Brilliant Blue [3]. However, there is public concern regarding the toxicity of synthetic food colourants, and some studies reported that these synthetic food blue colourants have side effects for humans, for example allergic reactions, hypertension, cerebral ischemia, cytotoxicity and genotoxicity [4]. Hence, for health reasons, these synthetic food blue colourants’ disadvantages prompt the urgent need to explore more blue pigment substitutes from natural products.

Some previous studies on blue natural colourants exist in the literature [3]. One study reported that bluish proanthocyanins named portsins were found in aged wines [5]. Although the researcher prepared its microparticle form to improve stability, the limitations of this colourant are the low rate of yielding and dyeing power [6]. Another natural blue colourant from colourless iridoid genipin needed chemical modifications [7]. Phycocyanin is a blue protein complex extracted from blue algae with multiple bioactivities. Blue extracts containing phycocyanin have been approved for specific food industries by the FDA and the European Union [3]. However, the high cost of the extraction and unstable properties of microorganisms and algae limit their wide application [8]. Natural indigo and marennine are also natural blue colourants, but environmental stability also restricts their comprehensive utilisation; for example, 40 °C would attenuate the colour of the natural indigo, and the bluish−green colour marennine has only been presented in an acid environment [9,10]. Therefore, finding natural blue colourants with low toxicity, cost, wide adaptability and robust stability is essential.

Anthocyanins (ACNs), a class of polyphenolic compounds which form the colour of flowers, fruits and vegetables, are widespread water−soluble natural colourants [11]. The colour of anthocyanins changes with environmental and molecular structure [12]. Anthocyanins present red colour at pH 1–2, purple colour at pH 6–7 and blue to yellow with an increase in OH^−^ [13]. According to an earlier study, blue anthocyanins were first found in *Clitoria ternatea* flowers named ternatin A1 [14]. The research on the living flower also indicated that the blue could be caused by metalloanthocyanins, intramolecular stacking and genetic modulation [15]. The blue effect of anthocyanins was studied by chelating with metal ions, and the results indicated that the blue hues of anthocyanins could be enhanced by metal [16]. Recent studies have reported that synthetic biology was leveraged to develop enzymatic transformation, which could turn the anthocyanins from red cabbages into blue [17]. Another study reported that the blue colour of ferric anthocyanins was improved in polysaccharides and gelatin [18]. However, these bluish anthocyanins existed with some defects. For example, adjusting the pH to change the blue colour of the anthocyanins required weak−alkaline environments, which may have led to difficulties in food production and stability. Natural bluish anthocyanins exhibit the problems of low content and complicated extraction, health issues from the addition of metal ions, and efficiency and cost concerns related to enzymatic transformation. *Lycium ruthenicum* Murr. (*LR*) is a long−lived shrub grown in salted and alkalised soil, which is also known as black goji, black fruit wolfberry, siyah goji, etc. The analysis of the chemical compounds demonstrated that these fruits were rich in anthocyanins, which were predominantly petunidin derivatives [16]. Based on the material foundation, *LR* has high potential for use as a natural food colourant.

Copigments are collections of compounds such as flavonoids, alkaloids, amino acids, organic acids, nucleotides, polysaccharides, metals or other anthocyanins that have the function of improving the colour of anthocyanins and their stability [19]. Existing studies have demonstrated that the addition of plant extracts as copigments can stabilise and enhance the colour of foods [20,21], and research suggested that multiple components of the juice substance work together in copigmentation, which enhances the colour appearance [22]. The occurrence of the copigment effect is usually accompanied by a red shift in λ_max_ and simultaneous bluing of the colour, producing a colour−enhancing effect [23]. This bluish−colour effect is mainly caused by the quinone bases and is stabilised by the copigments [24]. Based on this theory, it is possible to obtain the blue colour anthocyanins by copigmentation.

In the development process, the colour of a natural food colourant is an essential indicator. The main existing methods for evaluating colours are International Commission on Illumination (CIE) colour space and the UV−Vis spectrum [25,26,27]. CIE proposed the CIE Lab colour space for evaluating colour coordinate systems in 1976. The Lab colour model consists of three metrics, lightness (+ brighter, − darker), redness−greenness (+ redder, − greener) and yellowness−blueness (+ yellower, − bluer). In addition, UV−Vis spectroscopy can be used to study the intensity and nature of the colour by studying the complementary spectrum of the target colour. For example, the complementary wavelength of blue light is 580–600 nm. Hence, we searched for and evaluated blue food colourants based on these two techniques from *LR*. 

Natural blue food colourant is rare and has numerous problems, such as high cost, instability and complicated extraction, and metal chelates with anthocyanins, causing health risks. This is a major hurdle for the food industry’s use of ACN−based blue colourants in foods or beverages. Studies have demonstrated that the colour of *LR* can be shifted to blue by pH influence [28], which is related to the formation of the anion quinoidal base [29]. However, anthocyanins from neutral *LR* exhibited a purple colour and were more unstable than those in weak alkaline, which limits application as a blue food colourant. Thus, the objective of this study was to screen compounds from the common copigments that could improve the blue tones of anthocyanins and to investigate the effect of different copigments on the colour stability of anthocyanins in neutral species. Based on the copigmentation effect, a total 10 of common food additives and biological agents were chosen. The copigments obtained from the screening had effects on the colour and stability of the anthocyanins, and we hypothesised that this function is exerted through intermolecular hydrogen bonding and π−π stacking. This study widens the understanding of anthocyanin colour and offers a convenient, available and more stable way to use it as the colourant in a specific food domain.

## 2. Results and Discussion

Blue pigments are rare in nature and even rarer in edible form. Modern artificial blue pigments are linked to environmental and health hazards, such as attention deficit and hyperactivity disorders in children [30]. In the face of such risks, there is an increasing demand for natural colourants. Anthocyanins from *LR*, which are rich in petunidin derivatives, have received much attention and research for their unique violet−blue colour and wide colour ranges and have great potential as a natural blue colourant [16]. It has been shown that copigments can enhance the colour and stability of anthocyanins by protecting chromophores from hydration [31]. Moreover, copigments can increase the redshift of λ_max_ and the absorbance of anthocyanins [23]. Therefore, we envisage a strategy to design blue anthocyanins based on the wide colour ranges of *LR*’s anthocyanins by screening for copigments. 

### 2.1. Optimisation of Reaction Time and Concentration

Firstly, the reaction conditions that could cause the anthocyanin colour to be close to blue needed to be confirmed. The targeted colours were purplish−blue, purple and purplish−red, and the CIE Lab parameters of the three colours are (28.00, 59.16, −49.50), (29.78, 58.93, −36.49) and (32.04, 59.67, −22.59), respectively. The colour difference was calculated between the sample and the targeted colours. The optimised reaction time and concentration condition, which caused the closest colour to the targeted colour, were determined.

The effect of ACN concentration on the colour difference is shown in Figure 1C. With the increase in the concentration of ACNs, the colour difference demonstrated a trend of initial decreasing and then increasing, and the changing trend of colour differences with three targeted colours was consistent. The results demonstrated that the colour difference at 0.4 mg/mL was the smallest among the three groups; thus, 0.4 mg/mL was chosen as the optimum ACN concentration. With the increase in anthocyanin content, the copigmentation occurred with the increase in colour intensity [32], which consisted of our observation. In Figure 1A, by comparing with the targeted colour, lightness (L) presented a decreased trend, but redness−greenness (a) and blueness−yellowness (b) reached their peak and bottom at 0.4 mg/mL, respectively, which indicated that this concentration was closest to the target colour, and the lightness was appropriate.

The effect of ACN reaction time on the colour difference is shown in Figure 1D. The trend of colour difference with the increase in the reaction time was assessed in the concentration experiments. The colour difference of ACNs at 10−30 min is significantly different from other groups. The colour difference at 20 min was lower than that at 10 and 30 min. Compared to the purple colour, the colour difference at 20 min (43.81 ± 0.02) was slightly lower than the value at 10 (46.17 ± 0.04) and 30 min (44.06 ± 0.06). Hence, 20 min was chosen as the reaction time for the improved colour of ACNs. With the increase in time, the colour of the stored anthocyanins commonly showed a fading trend [33]. In Figure 1B, lightness increased, and redness−greenness and blueness−yellowness presented the opposite trend, which indicated the attenuation of the colour with the increase in reaction time, and the colour at 20 min was closest to the target colour.

### 2.2. Colour Improvement of the Mixed System

After determining the optimal conditions, the colour improvement of ACNs was further investigated. Pro−Xylane (PX), ectoin (ECT), ergothioneine (EGT), α−arbutin (α−ABT), β−arbutin (β−ABT), dipotassium glycyrrhizinate (DG), nicotinamide (NTM), ceramide (CRM), allantoin (ALT) and inositol (IST) were selected from common functional excipients. Furthermore, based on the structure and colour of neutral quinoidal base anthocyanins, the targeted colour was selected as Oxford Blue, the CIE Lab parameter was (12.88, 5.81, −27.03) and the effect of copigmentation on the colour differences of ACNs was explored. 

The effect of the concentration of compounds on colour differences in anthocyanin solution is shown in Figure A4 and Figure A5. In Figure A5C–G, compared with the control group, the colour difference did not change significantly with the increase in compound concentration. The result indicated that EGT, α−ABT, β−ABT, IST and CRM at 0–90 mg/mL could not turn ACNs blue. In Figure A4 and Figure A5A,B, the colour difference changed significantly with the increase in compound concentration. To be more specific, the addition of PX, ECT and DG demonstrated a significant decrease in the colour difference. The addition of ECT decreased the colour differences from 21.19 ± 0.18 at 0 mg/mL to 14.72 ± 0.12 at 90 mg/mL, and the colour difference in the DG group (20.38 ± 0.12) decreased to 11.89 ± 0.08 with the increase in concentration. According to our observations, the NTM group’s colour difference decreased significantly; however, the colour quickly faded. In the ALT group, the decrease in colour difference exhibited a non−significant trend, from 20.38 ± 0.03 to 17.35 ± 0.05 at 40 mg/mL, thus NTM and ALT were excluded from the study. The ACN binary complex with PX, ECT and DG can significantly reduce the colour difference with the targeted blue. Hence, the above three compounds were selected for further investigation.

The copigmentation effects are related to concentration [32]. Hence, the effects of these compounds’ concentrations on the colour of ACNs were continuously studied after the primary screening. These colour changes are presented as the CIE Lab parameters in Figure 2 and Table A1.

Regarding the CIE Lab parameter (Figure 2 and Table A1), with the increase in PX concentration, the Lab parameters of the complex decreased, and the lightness (L) decreased from 31.55 ± 1.83 to 16.61 ± 0.73 at 18 mg/mL concentration, while the redness−greenness (a) decreased from 53.64 ± 3.23 to 26.1 ± 1.56, and the yellowness−blueness (b) decreased from −35.52 ± 1.45 to −52.03 ± 1.32. The results demonstrate that the ACN’s colour shifted to green and blue, and the lightness decreased with the increase in PX concentration. The addition of ECT decreased the L value to the lowest (20.84 ± 1.32) at 140 mg/mL. It is indicated that ECT could decrease the b value and lead to an increase in the expression of blue colour. Adding DG (20 mg/mL) can improve the L value from 29.88 ± 2.14 to 47.36 ± 3.91. The increase in DG concentration had little effect on the b value but significantly decreased the a value, which decreased from 37.57 ± 2.16 to 30.21 ± 0.02. DG could enhance the ACN’s lightness and reduce the redness−greenness. According to the literature, the blue colour of the ferric anthocyanins in gelatin and polysaccharides improved, and the lightness declined with the gelatin concentration increase [18]. ECT is an aromatic acid compound with a planar structure of a pyrimidine−pyrrole ring. Common aromatic acid is usually observed as intermolecular and intramolecular copigment effects with anthocyanins, resulting in adjusting colour [34]. PX is a type of xylose in the colloid form. The addition affected the lightness, which agreed with the previous study, but the previous study was based on ferric anthocyanins, which naturally present the blue colour. Our study suggests that the PX directly improved the blue colour without metal ions. DG is used for potassium supplements and sweeteners for sports drinks, and the related research indicated that DG could enhance carotenoids’ photostability and antioxidant properties [35]. The chemical structure of DG has numerous conjugated double bonds, which could bring the λ_max_ redshift, forming a bluer colour.

#### Blue Contribution Calculation

The blue contribution is shown as the area under the curves in Figure A6 for ACNs, ACNs−PX, ACNs−ECT and ACNs−DG. The calculated contribution value of the blue colour is presented in Table 1. The ACNs had the smallest value with 13.62, and the next smallest was the ACNs−DG with 23.38. ACNs−PX had the highest value of 47.94, and ACNs−ECT had the next highest value (28.77). The result indicated that adding these compounds could significantly increase the blue contribution of the anthocyanin solution in the visible spectrum. The reason that the colour of ACNs−PX was close to Oxford Blue is that addition of the PX substantially facilitated the increased absorbance of the blue area in the visible spectrum and the decrease in the solution’s lightness, resulting in the declination of the values of a and b in CIE.

### 2.3. Accelerated Storage Tests

#### 2.3.1. Absorbance and Kinetics Degradation

The accelerated storage evaluated the stability of the optimised complexes. Firstly, the absorbance changes at the maximum absorbed wavelength were considered. The results are illustrated in Figure 3C. Compared to the anthocyanins group, adding the PX and the ECT significantly increased the absorbance to 3.05 ± 0.05 and 2.20 ± 0.07, and the DG group failed to express significant changes. In the absence of the compound’s addition, ACNs rapidly decreased from 1.32 ± 0.03 to 0.62 ± 0.04 within 3 h and then slowly decreased to 0.30 ± 0.01 within 3–168 h. However, it took 9 h for the ACNs−PX group to decrease to 0.65 ± 0.01 from 3.05 ± 0.02. As for the ACNs−DG group, the absorbance slightly decreased to 0.67 ± 0.02 from 1.38 ± 0.01, and this process lasted for 168 h. 

Then, the kinetics degradation of the accelerated storage was evaluated according to the absorbance data. According to the literature, the degradation kinetics of anthocyanins follow the zero−order and first−order kinetic models. The results are shown in Table 2. When ACNs were stored in the dark at 40 °C, the t_1/2_ was 2.84 ± 0.04 (zero−order kinetics) and 5.60 ± 0.15 h^−1^ (first−order kinetics). After adding the PX or DG, the t_1/2_ significantly increased to 6.90 ± 0.02 and 137.42 ± 0.63 h^−1^, respectively. According to the literature, neutral anthocyanins stored at 50 °C commonly presented a rapid colour loss of 50% within 4 h [36], and petunidin derivatives were more unstable than the cyanidin derivatives. Hence, the half−life of anthocyanins in our study was shorter than previously observed. The slight improvement of stability of ACNs−PX could be the reason for the presence of xylogalacturonan or acetylation at C−2 or C−3, which inhibited the third hydrogen bond [37]. Although ACNs−ECT improved the blue colour, it attenuated the stored stability, possibly indicating that the pyrimidine heterocycle had a pair of nitrogen atoms, which had a strong electron attraction effect. The k of the ACNs−DG was the lowest among the treated groups, indicating that the ACNs−DG mixed system had the most remarkable improvement in the stability of anthocyanins, which could be the reason for the intermolecular interactions such as van der Waals forces, hydrogen binding, hydrophobic forces and π−π interactions [24,38]. According to the literature, the stability improvement in this mixed system could be the conformation’s protection of the flavylium cation from water attack [39,40].

#### 2.3.2. Colour Analysis

The colour differences and colour appearances were explored during the accelerated storage. The swatch of the colour change of the anthocyanin complex is shown in Figure 3A. The ACNs turned light purple at 3 h and light yellow after 24 h. Interestingly, the ACNs−PX group remained deep blue at 0–3 h, gradually turned green at 6–24 h and browned after 24 h. The ACNs−DG group remained purple at 0–24 h. Compared with the ACNs group, the ACNs showed significantly increased blue tone with the addition of the ECT. However, faster degradation occurred within 3 h in the ECT treated group. The results indicated that the mixed system between ACNs and PX or ACNs and DG was beneficial in improving colour stability, and ACNs−PX possessed a potential blue tone.

The results of the colour differences are displayed in Figure 3B. The ACNs and ACNs−ECT groups rapidly changed within 3 h, and colour differences slightly increased with the storage time in an insignificant trend. These results were consistent with the colour swatch, which demonstrated that the colour of the two groups presented a quick fade in the accelerated storage, and ECT aggravated this situation. It should be noted that, after adding PX or DG, the changes in the colour differences tended to be gradual. To be more specific, the colour differences of ACNs−PX increased from 0.04 ± 0.00 to 92.74 ± 0.31 during 12 h storage, and ACNs−DG gradually increased from 0.05 ± 0.00 to 49.74 ± 0.11 for 168 h. The substantial changes in the colour differences of the ACNs−PX could be due to the broad shift of their colour gamut. Namely, the colour shifted from a deep blue tone to a deep yellow with the increase in storage time. From the results of colour differences, ACNs−DG significantly improved stability during storage and controlled the colour differences.

### 2.4. FTIR Analysis

The FTIR spectrums of mixed systems were characterised and are displayed in Figure 4A. ACNs belonged to the flavonoids. There were a few typical features of flavonoids in the infrared spectra: the O−H stretching vibration at 3420 cm^−1^, C−H stretching vibration at 2950 cm^−1^, C−O stretching vibration at 1600 cm^−1^ and C−C stretching vibration at 1550 cm^−1^ as well as the C−O−C bending vibration at 1025 and 980 cm^−1^ [41]. After complexing with the compounds, the complex spectrum exhibited differences compared to the ACNs. For the ACNs−PX, the wavenumber of the O−H stretching vibration decreased to 3220 from 3420 and 3350 cm^−1^, suggesting the formation of intermolecular hydrogen bonds [42]. As for the ACNs−ECT and ACNs−DG, the absorbance at 1715–1000 cm^−1^ decreased. Peaks at 1100–1000 cm^−1^ slightly shifted, exemplified by the wavenumber of C−O stretching vibration shifting from 1140 to 1130 cm^−1^ in ACNs−ECT. These phenomena could indicate the existence of molecular interaction between the carboxyl groups of the compounds and the hydroxyl groups of the ACNs. The FTIR results indicated hydrogen bond interactions existed in these complexes, and these interactions changed the spectroscopic properties of anthocyanins to some extent.

### 2.5. ^1^H NMR Analysis

The chemical shift of ^1^H NMR can be used to describe the surrounding environment of the nucleus, and the change in the electron environment around the nucleus causes the shielding or de−shielding effect on the H atom. To be more specific, the decrease or increase in the intensity of the electron cloud of the H atom is related to the change in chemical shifts to low fields (de−shielding effect) or high fields (shielding effect) [43]. The aim of NMR analysis is to assess structural sites and relevant information of intermolecular interaction using the change of chemical shifts. The primary pigment from *LR* was petunidin−3−*O*−coumaroylrutinoside−5−*O*−glucoside (p3cr5g) as a research object for exploring the interaction among the complex. The structural identification of the p3cr5g was based on the literature [44]. From Figure 4B−D, in ACNs−PX, the chemical shifts of p3cr5g were shifted towards high fields with the increase in the compound concentration; for example, 7.1 (H−6, s) of p3cr5g at control shifted to 7.26 (H−6, dd, J = 14.2 Hz) at 1:20 molar ratio (ACNs: PX), which suggested that the compounds interacted with the p3cr5g and increased the density of the electron cloud of the H atom at the core structure and acylated groups via hydrophobic π−π interaction. The same situation occurred in the other two groups. It was observed that H−4, H−6 and H−8 of p3cr5g occurred in the intramolecular interaction during the process of complexing [45]. As the concentration of the compound increases, the chemical shift of the compound also shifts downfield, such as chemical shifts in p3cr5g−PX, 9.09 (s), 8.83 (d, J = 5.8 Hz), 8.77 (dd, J = 8.1 Hz) and 8.02 (dd, J = 8.2, 5.8 Hz) were shifted to 8.85 (d, J = 2.5 Hz), 8.62 (dd, J = 5.0, 1.8 Hz), 8.20−8.15 (m) and 7.52 (dd, J = 7.9, 4.9 Hz), indicating that the chemical shift of the H atom of PX to the lower field was affected by the interaction between the complexes. ACN chromophores and glucose moiety can participate in this binding interaction with polysaccharides or hydroxyl groups of other compounds, which facilitated the stability [46]. To sum up, an interaction in the complex affected the H atom’s chemical shift, which occurred in the de−shielding and the shielding effect, and interaction common occurred in the core structure of ACNs.

### 2.6. AFM Characterisation

Atomic force microscopy (AFM) is a technique that can characterise microstructure and surface morphology by measuring the interaction force between probes and atoms [47]. AFM allows stable, high−resolution images of the sample surface to be obtained. The aggregation between anthocyanin complexes was evaluated by analysing various parameters of the surface morphology, which could facilitate exploring the differences of their stability at the molecular level. The AFM scan results of complexes are shown in Figure 5, and the surface heights are shown in Table 3. The surface height of p3cr5g was 0.92 ± 0.11 nm. After adding PX, the surface structure was more uniform, and the height was significantly increased to 2.13 ± 0.14 nm, which indicates that the molecular aggregation was increased, and the conformation was elevated. After adding ECT and DG, the surface heights were 1.23 ± 0.17 and 1.42 ± 0.04 nm, respectively, significantly higher than the p3cr5g. These results indicated that the compound and anthocyanins had intermolecular aggregation, but the structure was not uniform. The results demonstrated that PX, ECT and DG could combine with anthocyanin molecules to produce intermolecular force.

Surface roughness plays a vital role in studying biomolecular interactions [48]. The roughness between these complexes is presented in Table 3. The surface of p3cr5g was relatively uniform, with mean roughness (R_a_) and root mean square roughness (R_q_) values of 0.148 ± 0.001 and 0.204 ± 0.003 nm, respectively. After adding PX, the R_a_ and R_q_ increased to 0.382 ± 0.046 and 0.502 ± 0.072 nm, respectively, which were the highest compared with the other two groups. After adding ECT, R_a_ failed to display significant changes, while R_q_ was slightly increased to 0.238 ± 0.019 nm. P3cr5g−DG’s roughness was significantly improved with R_a_ and R_q_ values of 0.176 ± 0.013 and 0.271 ± 0.022 nm, respectively. According to the literature, the increase in surface roughness interacts with the compound molecules, and the aggregation of chains occurs, forming more protrusions and wrinkles, increasing the roughness [49].

### 2.7. Computational Analysis

Independent gradient model (IGM) analysis is an analytical method based on the superposition of initial densities in the free state of atoms to visually study covalent and non−covalent interactions. According to previous research [50], two tautomeric structures of the anthocyanins are presented in a neutral environment. Hence, the molecular structures of the complexes were initially established based on two species of neutral anthocyanins (Figure A7). The complexation energy of the complexes after the IGM analysis is presented in Table 4. The complexation energy of the p3cr5g−PX, p3cr5g−ECT and p3cr5g−DG in two species was (−27.21/28.60), (−19.03/19.91) and (−50.63/49.15) kcal/mol, respectively. The result indicated that the complexation energy of the p3cr5g−DG was the lowest and had the highest affinity, which was more stable than the other two complexes. The complexation energy of p3cr5g−PX was higher than p3cr5g−ECT, which indicated that the addition of the PX obtained higher stability than the ECT. The order of the complexation energies of the three compounds with anthocyanins from large to small was DG > PX > ECT. The addition of DG could significantly improve the complexation energies and stability. Furthermore, the calculation results of the complexation energy results agreed with the analysis of degradation kinetics of the complexes. We found that two species of anthocyanins in a neutral form could not affect the complexation’s combination according to the results of the calculation. It was speculated that the co−existence of anthocyanin isomers of neutral anthocyanins would not affect the stability of complexes.

The hydrogen bond and van der Waals forces played a vital role in analysing the molecular structure, stability and degradation dynamics. We utilised the IGM analysis to explore the interactions between multiple interacting areas using the index of δg^inter^. The δg^inter^ surface results are presented in Figure 6 (A_4′_ specie) and Figure A8 (A_7_ specie). The surface stands for the interaction areas of the hydrogen bond and van der Waals forces, and the blue area stands for the attraction force. The interaction sites of p3cr5g−PX and p3cr5g−ECT were concentrated on the anthocyanin chromophores, and there was less interaction on the glycosyl chain. At the same time, DG had a large molecular conformation that retained a broader interaction area with p3cr5g, including glycosyl chains. The attraction force results of p3cr5g−PX, p3cr5g−ECT and p3cr5g−DG indicated the interaction site at C−4, C−1 of the glycoside and C−2′ and C−3′, respectively. In conclusion, compared with p3cr5g−ECT and p3cr5g−PX, p3cr5g−DG has a wider non−covalent region and more mutual attraction force.

As shown in Figure 6 and Figure A8, the green iso−surface represents the action area of van der Waals force or π−π stacking. The redder the atoms, the more significant their contribution to intermolecular interactions. It can be observed that p3cr5g−DG had more red atoms in the interaction force. Hence, the result indicated that the binding ability of the p3cr5g−DG complex was better than p3cr5g−PX and p3cr5g−ECT. This result agreed with the analysis of the complexation energy. 

Hirshfeld surface analysis was used for the areas where the molecules were in contact. The result is shown in Figure 6, and areas with close interactions are red or white, dominated by hydrogen bonds, while blue areas represent weaker interactions or greater molecular distances. The results demonstrated a hydrogen bond between the hydroxyl group in the p3cr5g−PX, which was crucial for stabilising anthocyanins. According to the literature, the polyhydroxy structure of anthocyanins easily forms hydrogen bonds with polysaccharides and enhances the binding force. The p3cr5g−DG formed more hydrogen bonds than the other because of DG’s many hydroxyl groups and multiple hydrogen bond systems, which improved the stability of anthocyanins. The quantity of hydrogen bond binding sites or interaction force regions was in the order of p3cr5g−DG > p3cr5g−PX > p3cr5g−ECT. In conclusion, hydrogen bonds improved the anthocyanin complex stability, and the result was consistent with previous stability experiments. 

## 3. Materials and Methods

### 3.1. Materials

Ripe *LR* was collected from Dulan County in the Qinghai−Tibet Plateau (latitude 36°44′ N, longitude 96°43′ E, altitude 3000 m) during the mature period (September 2020). Prof Qingbo Gao of the Northwest Institute of Plateau Biology, Chinese Academy of Science, identified the herb. A voucher specimen (NWIPB−0334881) was deposited at the herbarium of Northwest Institute of Plateau Biology, Chinese Academy of Science. The primary pigment, petunidin−3−*O*−coumaroylrutinoside−5−*O*−glucoside (p3cr5g), was prepared with a semi−preparative HPLC system (NZ−7000, Hanbon, Huaian, China) from *LR* extracts, with purity > 95%, which was determined with an analytic DAD−HPLC system (1200, Agilent, Palo Alto, CA, USA). The HPLC spectrum and the data of ^1^H NMR and ^13^C NMR are presented in Figure A1, Figure A2 and Figure A3. PX, ECT, EGT, α−ABT, β−ABT, DG, NTM, CRM, ALT and IST were purchased from Xi’an ChaoBang Bioscience Co., Ltd. (Xi’an, China) with purity > 95%. AB−8 macroporous resins were purchased from Hebei BaoEn Biotechnology Co., Ltd. (Cangzhou, China).

### 3.2. Anthocyanin Extraction

Anthocyanins were extracted from dried fruits of *LR* with 70% ethanol (ethanol/fruit = 15:1 *v/w*) facilitated with microwaves. The extracts were loaded onto a column containing macroporous resins (AB−8) for enrichment. Polysaccharides were removed from pure water, followed by the elution of anthocyanins with 95% ethanol. After that, ethanol was eliminated from a rotary evaporator (N−1300, EYELA, Tokyo, Japan) at 55 °C, and powdered anthocyanins were obtained and stored at 4 °C. The extraction process of anthocyanins was performed in the dark.

### 3.3. Preparation of Stock Solutions

ACN stock solutions were prepared at a 10 mg/mL concentration. PX stock solutions were prepared at the concentration of 10 mg/mL. ECT, EGT, α−ABT, β−ABT, DG, NTM, CRM, ALT and IST stock solutions were 100 mg/mL. The solvent of the above solutions was pure water. 

### 3.4. Optimisation of Reaction Time and Concentration

Due to ACNs presenting purple colour in neutral forms, copigmentation can turn their colour to purplish−blue and purplish−red. Hence, the purple, purplish−blue and purplish−red CIE parameters were initially chosen to calculate the optimisation’s colour differences to explore the effect of reaction time and concentration on the solutions’ colour.

To optimise the concentration, anthocyanin stock solution was diluted with the pure water at the concentrations of 0.05, 0.10, 0.20, 0.40, 0.60, 0.80, 1.00, 2.00, 3.00 and 4.00 mg/mL, and the colour of the diluted ACNs was detected after storage in the dark for 20 min. To optimise the reaction time, the diluted ACNs were stored for 0, 10, 20, 30, 40, 50, 60, 70, 80 and 90 min in the dark at the optimised concentration. The colour of these above solutions was detected using a colourimeter (CS−820N, CHN SPEC, Hangzhou, China). 

### 3.5. Colour Improvement of the Mixed System

Blue natural pigments are uncommon, causing artificial blue pigments to be predominantly used in the food and dye industries. Hence, Oxford Blue, the traditional blue colour, was chosen as the targeted colour. The experiment aimed to screen out the mixed system, which could turn the colour of the ACN solution close to the targeted colour.

To explore the colour improvement of the mixed system, stock solutions of these compounds at different concentrations were diluted into the ACN solution at the optimised final concentration. A colourimeter and a UV−Vis spectrometer (T6, Persee, Beijing, China) detected the colour and the spectrum of the above solution. 

#### Blue Contribution Calculation

The blue colour contribution calculation from the literature was adopted with slight modifications [17]. The blue contribution region of the visible light spectrum was defined as the absorbance from 580 to 600 nm. The left Riemann sum was used for calculating the integration of these areas [51]. Based on the result, the absorbance was the main significant difference between groups, and the normalisation would erase this trend and concentrate on the changes of the λ_max_. However, the changes in λ_max_ were not significant. Hence, normalisation was not used for the data.

### 3.6. Accelerated Storage Tests

ACNs−PX, ACNs−ECT and ACNs−DG were plated and sealed in the centrifuge tube of 50 mL. The mixed systems contained 12 mg/mL PX, 0.4 mg/mL ACNs, 120 mg/mL ECT and 120 mg/mL DG. All tubes were stored at 40 °C in the dark for 168 h to accelerate the degradation of ACNs [43]. 

#### 3.6.1. Measurement of Colour

The colourimeter determined the sample solution. The colourimeter was set at a 10° observer angle with D65 standard illuminant. CIE parameters of lightness (L), redness−greenness (a) and yellowness−blueness (b) were measured. The colour swatch was visualised with GraphPad Prism 8.0 (GraphPad Software Inc., San Diego, CA, USA) according to the colorimetric data of each sample. Because the L* a* b* system alone cannot directly indicate the colour changes, total colour differences (ΔE) were used, which were calculated with Equation (1):(1)$$∆E=\sqrt{{\left(L-L_{0}\right)}^{2}+{\left(a-a_{0}\right)}^{2}+{\left(b-b_{0}\right)}^{2}},$$
where L_0_, a_0_ and b_0_ represent the initial values.

#### 3.6.2. Measurement of Absorbance

The chemical stability of ACNs in the accelerated was determined by measuring the decrease in absorbance at their maximum absorbance by using a UV−Vis spectrophotometer.

#### 3.6.3. Degradation Kinetics

ACN’s degradation kinetics were analysed by zero−order and first−order models in the solution conditions. The zero−order kinetics were calculated as Equations (2) and (3), and the first−order kinetics were calculated as Equations (4) and (5).
(2)$$A_{t}=A_{0}-\mathrm{kt},$$
(3)$$t_{1/2}=\frac{A_{0}}{2k},$$
(4)$$\ln\frac{A_{t}}{A_{0}}=-\mathrm{kt},$$
(5)$$t_{1/2}=\frac{\ln2}{k},$$
where A_0_ is the initial maximum absorbance, A_t_ is the maximum absorbance at t hours of storage and k is the degradation rate constant.

### 3.7. FTIR Analysis

The interaction between different molecules was studied using infrared spectroscopy. The FTIR spectra of ACNs, PX, ECT, DG, ACNs−PX, ACNs−ECT and ACNs−DG were characterised by the FTIR system with a single−reflection attenuated total reflectance (ATR) accessory (Nicolet iS10, Thermo Fisher Scientific, Waltham, MA, USA). Before the samples of ACNs, ACNs−PX, ACNs−ECT and ACNs−DG were freeze−dried, the binary solutions contained the ACNs (0.4 mg/mL), PX (12 mg/mL), ECT (120 mg/mL) and DG (120 mg/mL). FTIR system scanned the samples in the 4000 to 500 cm^−1^ range at the resolution of 8 cm^−1^ with 64 scans.

### 3.8. ^1^H NMR Analysis

The binary complex solution was pre−prepared, containing the optimised compound and p3cr5g with the molar ratios of 20:1, 10:1, 5:1 and 2:1. After the equilibration of the mixed system, the solutions were freeze−dried using a lyophiliser (10C, Foring, Beijing, China). Before NMR analysis, samples were dissolved in deuterated water (D_2_O, Merck, Darmstadt, Germany) and equilibrated for 30 min. The ^1^H NMR analysis was performed on the NMR spectrometer (AV 600MHz, Bruker, Billerica, MA, USA). The chemical shifts were recorded and analysed.

### 3.9. AFM Characterisation

The binary complex solution was freeze−dried and contained the optimised compound and p3cr5g with a molar ratio of 10:1. The dried sample was dissolved with pure water, and 1 μL of the dissolved sample was spread on the freshly cleaved mica sheet (1.8 cm^2^). Then, scanning with an AFM system (Dimension Icon, Bruker, MA, USA) at the scan rate of 1.0 Hz in non−contact mode after volatilising the solution at room temperature. Average roughness (R_a_) and root mean square roughness (R_q_) were measured. All experiments were performed in triplicate at 25 °C and 25%RH, and the same probe was used for analysis.

### 3.10. Computational Chemistry Analysis

The 3D conformation of PX and ECT were downloaded from the PubChem databases with accession IDs 16666733 and 126041, respectively. The structures of DG and p3cr5g in neutral species were drawn in Chem3D and geometry optimised by MM^2+^ for the minimum energy conformation. More than 100 initial structures of different binary complexes (ACNs−PX, ACNs−ECT, ACNs−DG) were generated by the Molclus program [52], and they were adopted as the initial structures for semiempirical quantum mechanical optimisation at the PM6−DH3 level using MOPAC2016. Ten different geometrical structures with low energy were screened out by Molclus and calculated by the DFT at the B3LYP/6−31G(d) level in the Gaussian 16 program [53]. The interaction energy (ΔE_int_) was calculated after BSSE correction. The independent gradient model analysis and Hirshfeld surface analysis were used to investigate the non−covalent interactions in the Multiwfn program [54,55].

### 3.11. Statistical Analysis

Statistical analysis was conducted using GraphPad Prism 8.0 software (GraphPad Software Inc., San Diego, CA, USA) and Origin 2018 (OriginLab Corporation, Northampton, MA, USA). All experiments were carried out in duplicate. One−way ANOVA (two−tailed, α = 0.05) was conducted to compare the differences among each group. A post hoc Tukey’s test with familywise α = 0.05 was performed in case of significance.

## 4. Conclusions

Natural blue pigment is rare. To make anthocyanins from *LR* retain their colour diversity and explore their potential as blue pigments, this study aimed to optimise food additives and biological agents and their complexation with anthocyanins to improve their colour and stability. The colour difference method evaluated the colour−improving effect of 10 different compound concentrations on ACNs. From the optimised experiments, PX, ECT and DG are three compounds that can shift the colour of ACNs to blue, among which the ACNs−PX complex is the closest to blue. Subsequently, the stability of the optimised complexes in the accelerated storage was evaluated with UV−Vis, CIE colour space and antioxidant ability in vitro. It was found that ACNs−DG could significantly improve the stability of anthocyanins with the lowest degradation rate, and ACNs−ECT accelerated the browning. The addition of PX helped to increase the stability of antioxidant ability. The stability mechanism was explored using IR, NMR, AFM and computational chemistry methods. The results of AFM demonstrated that the surfaces of the optimised complexes could be combined to form bumps and wrinkles, among which DG was the most stable combination. The analysis found that non−covalent forces such as hydrogen bonds and van der Waals forces were the source of their stability, as assessed using IR, NMR and computational chemistry.

In summary, this study found three optimised food additives and biological agents which could improve the blue colour and stability of ACNs. In this application, we can determine which additive (PX or DG) to use according to the purpose of blue colour or stability. These regular patterns could enhance the anthocyanins’ colour stability and offer a new perspective on preparing natural blue pigment from anthocyanins.

## Figures and Tables

**Figure 1 molecules-27-06089-f001:**
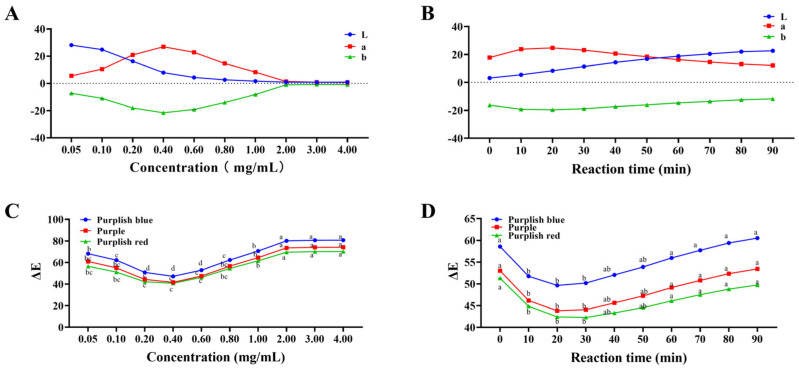
The optimisation of concentrations and reaction time of anthocyanin colour. (**A**) The effect of concentrations of anthocyanins on International Commission on Illumination (CIE) parameters; (**B**) the effect of reaction time of anthocyanins on CIE parameters; (**C**) the effect of concentrations of anthocyanins on colour differences; (**D**) the effect of reaction time of anthocyanins on colour differences. Groups with different letters indicate significant differences, *p <* 0.05; the same letters indicate non−significant differences, *p >* 0.05.

**Figure 2 molecules-27-06089-f002:**
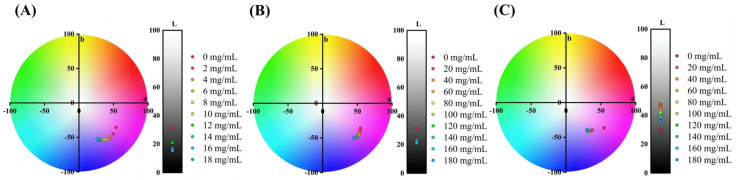
The effect of concentration of compounds on the colour appearance of anthocyanins. (**A**) Anthocyanins−Pro−Xylane (ACNs−PX); (**B**) Anthocyanins−Ectoine (ACNs−ECT); (**C**) Anthocyanins−Dipotassium glycyrrhizinate (ACNs−DG).

**Figure 3 molecules-27-06089-f003:**
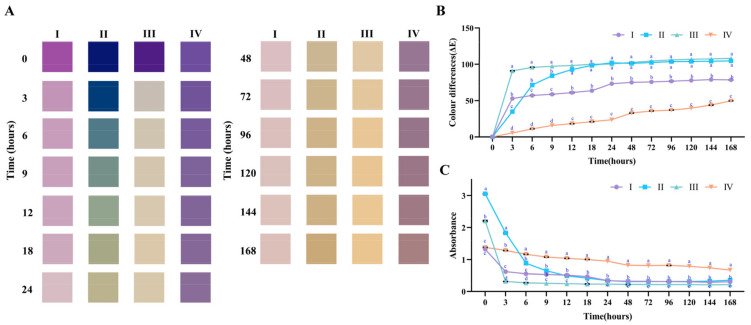
The stability evaluation of anthocyanin complex at 40 °C storage within 168 h. (**A**). The colour swatch; (**B**) the colour changes; (**C**) the absorbance changes; I: ACNs (0.4 mg/mL); II: ACNs (0.4 mg/mL)−PX (12.0 mg/mL); III: ACNs (0.4 mg/mL)−ECT (120.0 mg/mL); IV: ACNs (0.4 mg/mL)−DG (120.0 mg/mL). Groups with different letters indicate significant differences, *p <* 0.05; the same letters indicate non−significant differences, *p >* 0.05.

**Figure 4 molecules-27-06089-f004:**
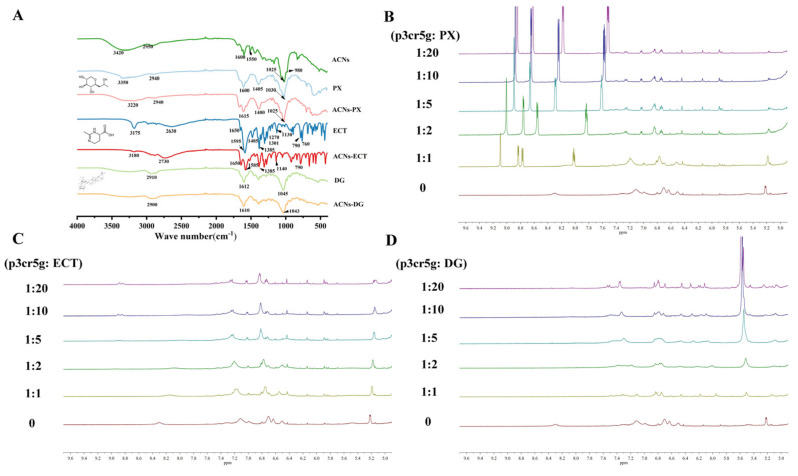
The spectrum of anthocyanin complex. (**A**) FTIR spectra of anthocyanin complex; (**B**) ^1^H NMR spectra of petunidin−3−*O*−coumaroylrutinoside−5−*O*−glucoside−Pro−Xylane (p3cr5g−PX) at different molar ratios; (**C**) ^1^H NMR spectra of petunidin−3−*O*−coumaroylrutinoside−5−*O*−glucoside−Ectoine (p3cr5g−ECT) at different molar ratios; (**D**) ^1^H NMR spectra of petunidin−3−*O*−coumaroylrutinoside−5−*O*−glucoside−Dipotassium glycyrrhizinate (p3cr5g−DG) at different molar ratios.

**Figure 5 molecules-27-06089-f005:**
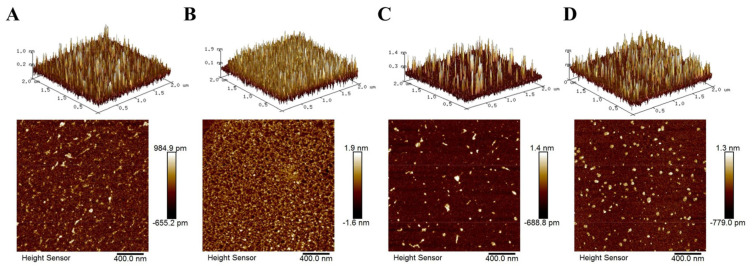
Atomic force microscope topography of anthocyanin compound−binary complex. (**A**) p3cr5g; (**B**) p3cr5g−PX; (**C**) p3cr5g−ECT; (**D**) p3cr5g−DG; three−dimensional images (upper) and two−dimensional images (lower) are presented.

**Figure 6 molecules-27-06089-f006:**
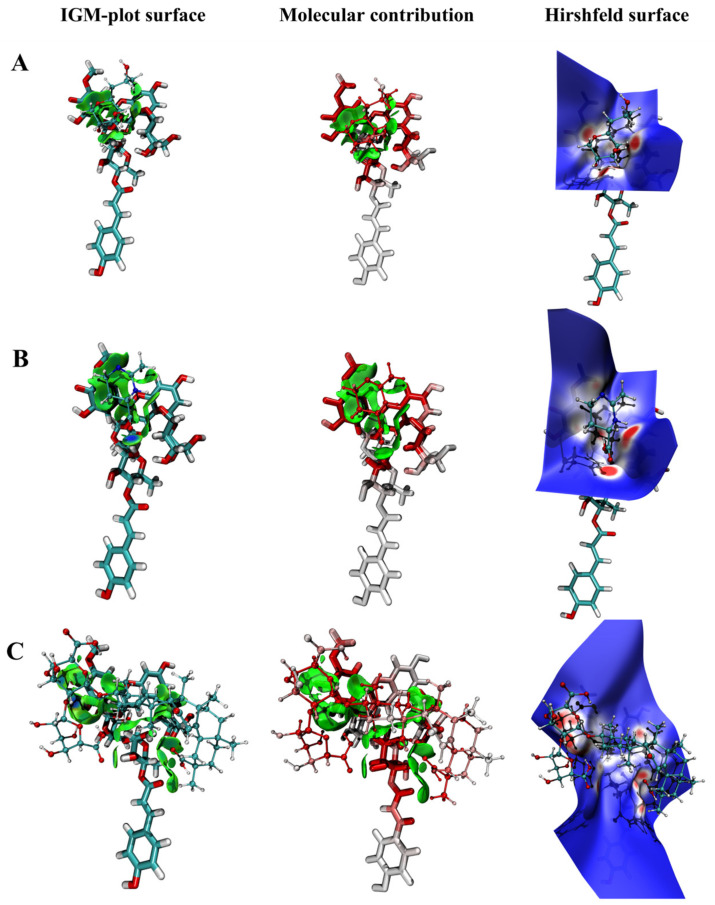
Non−covalent interaction for the anthocyanin (A_4′_ species) compound−binary complex by IGMH analysis. (**A**) p3cr5g−PX; (**B**) p3cr5g−ECT; (**C**) p3cr5g−DG.

**Table 1 molecules-27-06089-t001:** The λ_max_ and the calculated blue contribution of the anthocyanin complexes.

Complexes	ACNs	ACNs−PX	ACNs−ECT	ACNs−DG
λ_max_ (nm)	540	560	550	570
Blue contribution	13.62	47.94	28.77	23.38

**Table 2 molecules-27-06089-t002:** Degradation kinetic parameters of anthocyanin complex.

Complexes	Kinetics Order	k (Hours^−1^)	t_1/2_ (Hours^−1^)
ACNs	Zero−order	0.23 ± 0.01 b	2.84 ± 0.04 d
	First−order	0.12 ± 0.00 c	5.60 ± 0.15 c
ACNs−PX	First−order	0.10 ± 0.01 d	6.90 ± 0.02 b
ACNs−ECT	First−order	0.65 ± 0.03 a	1.07 ± 0.05 e
ACNs−DG	First−order	0.0051 ± 0.00 e	137.42 ± 0.63 a

Note: Groups with different letters indicate significant differences, *p <* 0.05; the same letters indicate non−significant differences, *p >* 0.05.

**Table 3 molecules-27-06089-t003:** Surface height and roughness of p3cr5g compound−binary complex.

Complexes	Surface Height (nm)	Average Roughness (nm)	Mean Square Roughness (nm)
p3cr5g	0.92 ± 0.11 c	0.148 ± 0.001 c	0.204 ± 0.003 c
p3cr5g−PX	2.13 ± 0.14 a	0.382 ± 0.046 a	0.502 ± 0.072 a
p3cr5g−ECT	1.23 ± 0.17 bc	0.148 ± 0.010 c	0.238 ± 0.019 bc
p3cr5g−DG	1.42 ± 0.04 b	0.176 ± 0.013 b	0.271 ± 0.022 b

Note: Groups with different letters indicate significant differences, *p <* 0.05; the same letters indicate non−significant differences, *p >* 0.05.

**Table 4 molecules-27-06089-t004:** Calculated quantum chemical parameters of p3cr5g compound−binary complex.

Complexes	Complexation Energy (kcal/mol)	Complexes	Complexation Energy (kcal/mol)
p3cr5g(A_4′_)−PX	−27.21	p3cr5g(A_7_)−PX	−28.60
p3cr5g(A_4′_)−ECT	−19.03	p3cr5g(A_7_)−ECT	−19.91
p3cr5g(A_4′_)−DG	−50.63	p3cr5g(A_7_)−DG	−49.15

## Data Availability

The data presented in this study are available on request from the corresponding author (pending privacy and ethical considerations).

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
