# Peer review of "Improved Stability of Blue Colour of Anthocyanins from Lycium ruthenicum Murr. Based on Copigmentation"

_molecules, 2022, doi:10.3390/molecules27186089_

Round 1

Reviewer 1 Report

The article “Improved stability of blue colour of anthocyanins from Lycium ruthenicum Murr. based on copigmentation” shows o use of copgmentation to better the stability of blue color natural anthocyanins. The findings of this article show a new perspective on the use of anthocyanins as a blue dye and contribute to its large-scale application in the food industry since blue dyes in nature are rare and synthetic dyes are used in foods and can have adverse effects on human health. However, I have some observations:

1) Please increase the introduction section and add a more detailed paragraph about pigmentation.

 2) Please suggest a perspective for using anthocyanins from Lycium ruthenicum Murr based on pigmentation and  the side effects of synthetic dyes in the Results and Discussions section.

Author Response

Response to Reviewer 1 Comments

Point 1: Please increase the introduction section and add a more detailed paragraph about pigmentation.

Response 1: We gratefully appreciate your comment. We added a more detailed paragraph for the introduction.

Introduction, Line 72-82, Adding: “Copigments are collections of compounds such as flavonoids, alkaloids, amino acids, organic acids, nucleotides, polysaccharides, metals or other anthocyanins that have the function of improving the colour of anthocyanins and their stability [19]. Existing studies have shown that the addition of plant extracts as copigments can stabilise and enhance the colour of foods [20,21], and research suggested that multiple components of the juice substance work together in copigmentation, which enhances the colour appearance [22]. The occurrence of the copigment effect is usually accompanied by a red shift in λmax and simultaneous bluing of the colour, producing a colour-enhancing effect [23]. This bluish-colour effect is mainly caused by the quinone bases and is stabilised by the copigments [24]. Based on this theory, it is possible to obtain the blue colour anthocyanins by copigmentation.”

Point 2: Please suggest a perspective for using anthocyanins from Lycium ruthenicum Murr based on pigmentation and the side effects of synthetic dyes in the Results and Discussions section.

Response 2: We sincerely appreciate your precious comment. We added a more detailed paragraph in the Results and Discussions section.

Results and Discussions, Line 110-120, Adding: “Blue pigments are rare in nature and even rarer in edible form. Modern artificial blue pigments are linked to environmental and health hazards, such as attention deficit and hyperactivity disorders in children [30]. In the face of such risks, there is an increasing demand for natural colourants. Anthocyanins from LR, which are rich in petunidin derivatives, have received much attention and research for their unique violet-blue colour and wide colour ranges and have great potential as a natural blue colourant [16]. It has been shown that copigments can enhance the colour and stability of anthocyanins by protecting chromophores from hydration [31]. Moreover, copigments can increase the redshift of λmax and the absorbance of anthocyanins [23]. Therefore, we envisage a strategy to design blue anthocyanins based on the wide colour ranges of LR’s anthocyanins by screening for copigments.”

Reviewer 2 Report

Abstract

L14         Erase ‘anthocyanins’

L14         CIE and AFM are unfamiliar abbreviations. Explain them when first mentioned.

L15         Release ACNs from the parenthesis

L17         ‘close’ rather than ‘closed’

L19         Remove ‘significantly’

Language has to be strongly improved throughout the manuscript!!

Introduction

Mention also some of the common names of the plant species investigated, e.g. black fruit wolfberry, siyah goji, and kaoke.

CIE lab is extensively used in this work. Include a description of what is measured and how blue colours are related to the measurements. This has been done in short as explained in line 179-180 and in the supplementary figure A6.

Results and Discussion

L97         Figure A3-C: If the figure is actively discussed in the section, it should be included in the manuscript rather than in the supplementary section.

L101-102              Not clear

Fig 1       The font size of the concentrations should be enlarged.

L123       Explain the abbreviations at the first time

L125       Neutral? Are we talking about pH?

L147       Rather than using a sentence as reference, simply write (Table A1).

L273       I really don’t see the purpose of 1H-NMR, and the fact that there is an upfield shift in the proton resonances when increasing the ratio of PX to CAN, what does it tell? Remember that by increasing PX you also make changes of the solvent. The explanation is unclear.

L279       p3cr5g: Write out the first time this occur in the text (petunidin-3-O-coumaroylrutinoside-5- 383 O-glucoside)!

L292       This paragraph is unclear. As with H NMR the rationale of using this technique is not well described.

L308       Unclear

L322       Write out full name of IGM at the first time for this abbreviation.

L324       ‘are’ rather than ‘were’?

L385       ‘with purity>95%’ How was this measured?

L393      ‘macroporous resins (AB-8)’ Supplier/producer?

L399       What solvent?

L427       ‘The left Riemann sum was used for calculating the integration of these areas.’ Include a reference.

L435       The experiments have been run at 40 ℃, whereas the literature references used 50 ℃ (L207). Why were the lower temperature chosen?

L448       Which UV–Vis spectrophotometer? Manufacturer?

Round 2

Reviewer 2 Report

The revised manuscript includes changes in accordance to the previous review.